# Cell-specific non-canonical amino acid labelling identifies changes in the de novo proteome during memory formation

Harrison Tudor Evans, Liviu-Gabriel Bodea*, Jürgen Götz*

Clem Jones Centre for Ageing Dementia Research, Queensland Brain Institute, The University of Queensland, Brisbane, Australia

**Abstract** The formation of spatial long-term memory (LTM) requires the de novo synthesis of distinct sets of proteins; however, a non-biased examination of the de novo proteome in this process is lacking. Here, we generated a novel mouse strain, which enables cell-type-specific labelling of newly synthesised proteins with non-canonical amino acids (NCAAs) by genetically restricting the expression of the mutant tRNA synthetase, NLL-MetRS, to hippocampal neurons. By combining this labelling technique with an accelerated version of the active place avoidance task and bio-orthogonal non-canonical amino acid tagging (BONCAT) followed by SWATH quantitative mass spectrometry, we identified 156 proteins that were altered in synthesis in hippocampal neurons during spatial memory formation. In addition to observing increased synthesis of known proteins important in memory-related processes, such as glutamate receptor recycling, we also identified altered synthesis of proteins associated with mRNA splicing as a potential mechanism involved in spatial LTM formation.

*For correspondence:
l.bodea@uq.edu.au (L-GB);
j.goetz@uq.edu.au (JG)

**Competing interests:** The authors declare that no competing interests exist.

## Introduction

Neurons are highly complex, compartmentalized cells that respond to a wide range of physiological and pathological signals by regulated changes to their transcriptome and proteome. A critical component in this process is the synthesis of new proteins, both in the cell body and at the synapse (*Davis and Squire, 1984*; *Hafner et al., 2019*; *Li and Götz, 2017*; *Schanzenbächer et al., 2016*). New protein synthesis is required for the formation, retrieval, and updating of long-term memories (LTMs), distinguishing this process from short-term memory (*Costa-Mattioli et al., 2009*; *Jarome and Helmstetter, 2014*; *Lopez et al., 2015*). One form of LTM is spatial LTM, which is used by organisms to recall spatial information about the environment, and is formed through the process of spatial LTM consolidation (*Squire et al., 2015*). During this process, neurons undergo a series of changes in different brain regions, with the hippocampus being identified as one of the most important brain structures in both spatial learning and LTM (*Clopath, 2012*; *Mayford et al., 2012*; *Ziegler et al., 2015*). Furthermore, inhibition of protein synthesis in the hippocampus, especially within the first 24 hr following spatial learning, has been demonstrated to prevent the formation of spatial LTM (*Alkon et al., 2005*; *Inda et al., 2005*; *Jarome and Helmstetter, 2014*). Therefore, identifying precisely which proteins need to be newly synthesised during spatial LTM consolidation is crucial in dissecting the underlying molecular mechanisms.

Several proteins and pathways have been linked to spatial LTM formation, as shown by candidate-based studies in which rodents were subjected to behavioural paradigms, such as the active place avoidance (APA) or Morris water maze test in order to induce spatial LTM formation (*Merlo et al., 2015*; *Paul et al., 2009*; *Plath et al., 2006*). However, a non-biased proteomic analysis of de novo protein synthesis during the formation of spatial LTM has not been previously achievable, because de novo synthesised proteins are chemically indistinguishable from those that are already

present in the cell. This limitation can, however, be overcome through non-canonical amino acid (NCAA) labelling of newly synthesised proteins. In this technique, NCAAs can be administered for a given period, during which they are integrated into the nascent polypeptide chain (*Figure 1A*) (*Hinz et al., 2013*). Unlike most other de novo protein tagging techniques, NCAA incorporation enables newly synthesised proteins to be visualised using fluorescent non-canonical amino acid tagging (FUNCAT), or to be purified using bio-orthogonal non-canonical amino acid tagging (BONCAT) (*Figure 1B*) (*Hinz et al., 2013*). This is achieved by reacting the azide group of the NCAA with a dibenzocyclooctyne (DIBO)-bearing tag, using strain-promoted azide-alkyne cycloaddition (*Figure 1B*) (*Beatty et al., 2010*).

Two widely used NCAAs are the methionine surrogates azidohomoalanine (AHA) and azidonorleucine (ANL). In AHA labelling, the endogenous translational machinery is used to tag newly synthesised proteins with AHA (*Figure 1A*) (*Dieterich et al., 2006*; *Hinz et al., 2013*; *Ullrich et al., 2014*). ANL, on the other hand, allows for cell-type-specific NCAA labelling, as it is not recognized by the eukaryotic methionine tRNA synthetase and therefore does not natively integrate into de novo synthesised proteins (*Link et al., 2006*). ANL instead requires the presence of a mutant tRNA synthetase, such as NLL-MetRS, for its incorporation (*Figure 2A*) (*Ngo et al., 2013*). Genetically restricting NLL-MetRS expression enables cell-type- or tissue-specific incorporation of ANL. Both AHA and ANL labelling have been used in a wide range of cell-types, tissues and organisms (*Alvarez-Castelao et al., 2019*; *Alvarez-Castelao et al., 2017*; *Erdmann et al., 2015*; *Liang et al., 2014*; *Lopez et al., 2015*; *McClatchy et al., 2015*; *Ullrich et al., 2014*). However, even though AHA and ANL labelling display an enhanced versatility compared to other de novo protein labelling techniques, such as stable isotope labelling with amino acids in cell culture (SILAC) or puromycin labelling, they are yet to be more widely used to examine the de novo proteomic changes which occur during complex rodent behaviour.

Here, we detail the use of NCAA labelling to identify changes in de novo protein synthesis which occur during the formation of spatial LTM in mice. For this, we generated the RC3 mouse strain, which enables Cre recombinase-dependent targeting of NLL-MetRS expression for cell-type- or tissue-specific incorporation of ANL. Upon crossing the RC3 strain with Camk2a-Cre mice, we successfully restricted ANL labelling of newly synthesised proteins to hippocampal neurons. We combined this labelling technique with SWATH-MS (sequential window acquisition of all theoretical fragment ions mass spectrometry) to examine how the hippocampal de novo proteome is altered in mice following spatial LTM formation. Our results confirmed a number of previously established memory-related proteins to be altered in synthesis during spatial LTM formation, and also highlight potential novel memory mechanisms, such as alterations in mRNA splicing.

## Results

### AHA labelling reveals increased hippocampal protein synthesis during spatial LTM formation

Given the well-established role of de novo protein synthesis in the process of spatial LTM consolidation (*Davis and Squire, 1984*), we sought to use NCAA labelling in combination with an unbiased proteomic analysis to identify which proteins were altered in synthesis during spatial LTM formation in mice, using a simple and robust behavioural paradigm.

To achieve this, we subjected mice to an accelerated version of the active place avoidance (APA) test, referred to as '30 min APA'. Mice were trained over 30 min to use spatial cues to avoid entering a designated shock zone (*Figure 1C*). Wild-type mice undertaking this task demonstrated spatial learning as revealed by their ability to significantly decrease the number of entries into the shock zone over the training period (*Figure 1C*). We also found that the mice were able to recall the location of the shock zone in a probe trial 24 hr later (*Figure 1C*), indicative of spatial LTM formation (*Plath et al., 2006*).

Having established this behavioural paradigm, we next sought to use AHA labelling to examine brain-wide de novo protein synthesis during spatial LTM formation. We have previously demonstrated that optimal AHA labelling following intraperitoneal (i.p.) injection occurs over a time-frame of 16 hr at a concentration of 50 μg AHA/gram body weight (gbw) (*Evans et al., 2019*). We therefore first confirmed that mice treated with 50 μg AHA/gbw immediately after training performed

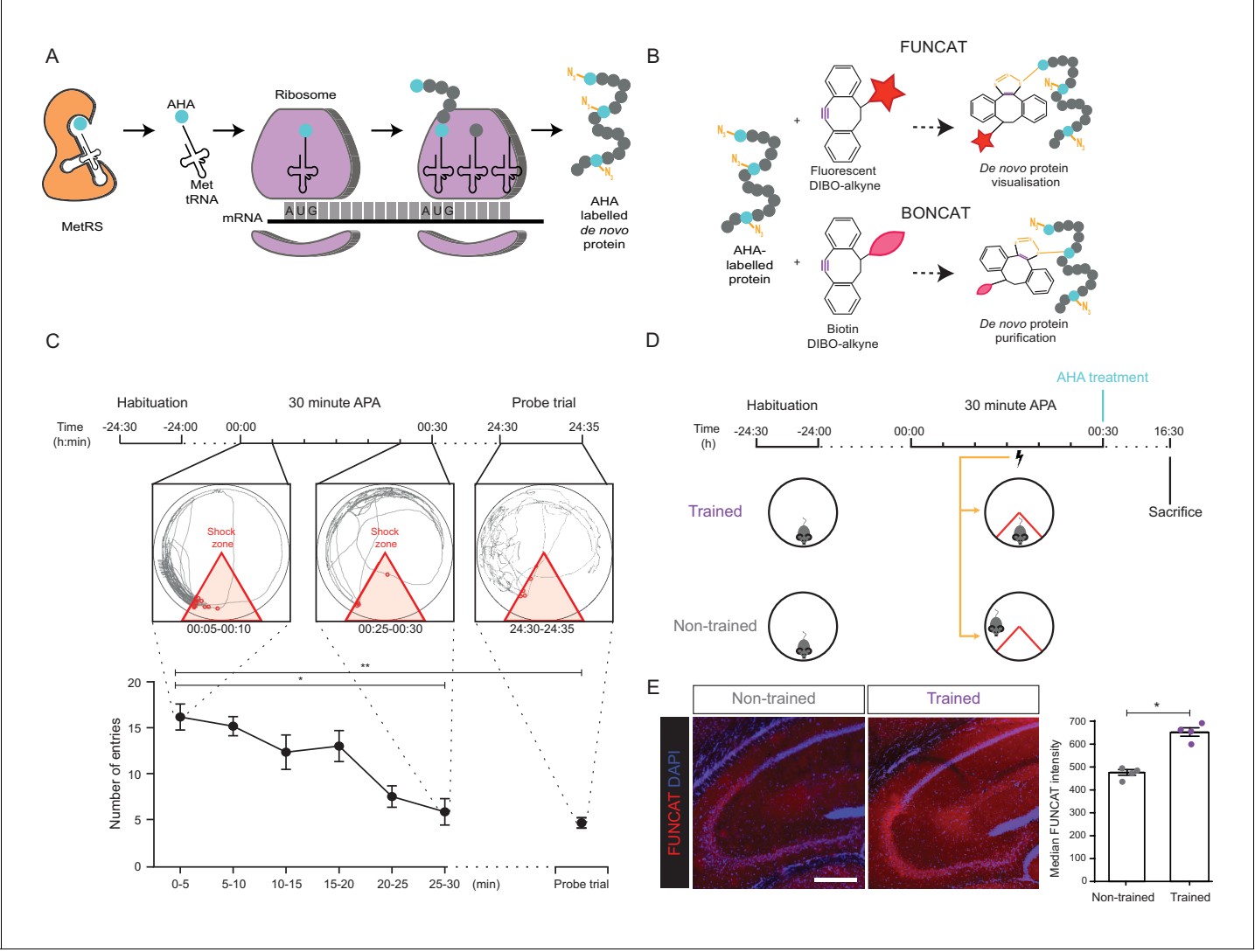

**Figure 1.** Long-term memory formation following training using the 30 min APA paradigm results in increased hippocampal de novo protein synthesis. (A) AHA is recognised by mouse methionine tRNA synthetase, MetRS (*Mars1*), and labels de novo synthesised proteins at amino-terminal and internal methionine residues using the endogenous translational machinery. (B) NCAA-labelled proteins can be covalently bonded to various tags through reaction of the azide group (orange) of the NCAA with the alkyne group (purple) of the tag. This enables NCAA-labelled proteins to either be visualised using fluorescent non-canonical amino acid tagging (FUNCAT) or to be purified using bio-orthogonal non-canonical amino acid tagging (BONCAT). (C) The 30 min APA paradigm results in spatial long-term memory formation. Mice trained over 30 min learned to avoid a designated shock zone (red), with significantly fewer entries into the shock zone being recorded between 25–30 min compared to between 0–5 min. In a 5 min probe trial held 24 hr after training, mice continued to avoid entering the shock zone, even in the absence of shocks, indicative of the formation of spatial LTM (n = 6 mice, one-way ANOVA, Dunnett's MCT, *p≤0.05, **p≤0.01). (D) Scheme of 30 min APA task for trained and non-trained mice. Trained mice received foot shocks upon entry into the designated shock zone, while non-trained mice received foot shocks at the same time as their trained partner and were therefore unable to undergo spatial LTM formation. Upon completion of the 30 min APA, mice were administered AHA and were perfused 16 hr later without undergoing a probe trial. (E) A significant increase in protein synthesis was observed in the hippocampus of trained compared to non-trained mice using FUNCAT (n = 4 mice, three sections per mouse, Student's paired t-test, *p≤0.05). Scale bar = 400 μm.
The online version of this article includes the following source data and figure supplement(s) for figure 1:

**Source data 1.** WT 30 minute APA behavioural data.
**Figure supplement 1.** AHA treatment does not inhibit spatial LTM formation.
**Figure supplement 1—source data 1.** AHA treated 30 minute APA behavioural data.
**Figure supplement 2.** Trained and non-trained mice show similar levels of stress, as measured by plasma corticosterone levels.

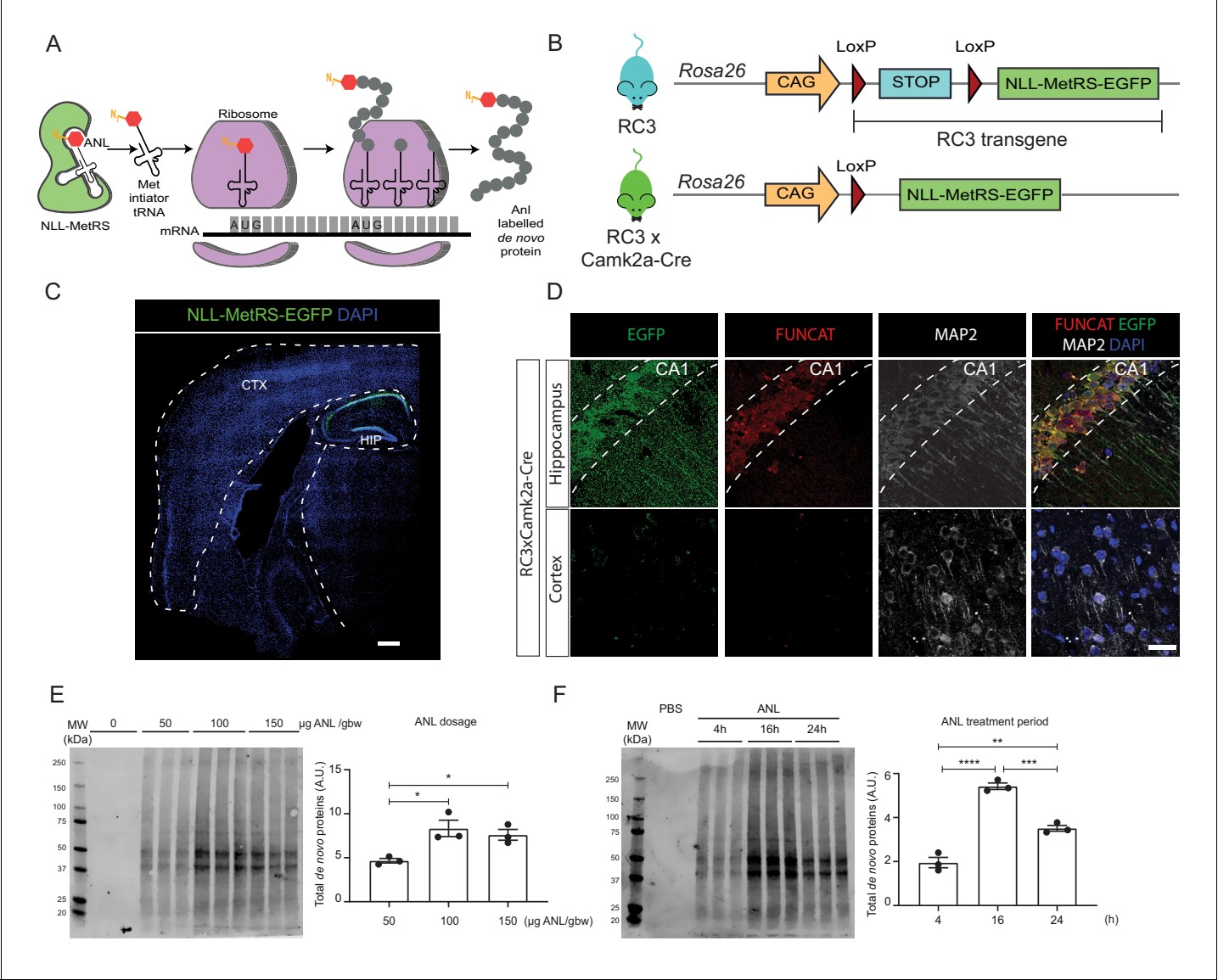

**Figure 2.** Rosa26 Cre Click-chemistry (RC3) mice enable cell type-specific labelling of newly synthesised proteins. (**A**) Incorporation of ANL requires the expression of NLL-MetRS, which allows this NCAA to be incorporated at the amino-terminus as proteins are being synthesized. (**B**) In RC3 mice, the expression of NLL-MetRS-EGFP is prevented by an upstream FLOX-STOP cassette. In RC3xCamk2a-Cre mice, Cre-recombinase is expressed in the hippocampus, resulting in the excision of the FLOX-STOP cassette from the RC3 transgene. This enables expression of NLL-MetRS-EGFP restricted to hippocampal neurons. (**C**) Immunohistochemical analysis confirmed Cre-specificity of NLL-MetRS-EGFP expression in RC3xCamk2a-Cre mice, with the EGFP signal being confined to neurons in the CA1, CA2, CA2 and dentate gyrus regions of the hippocampus. Scale bar = 400 μm (**D**) FUNCAT staining confirms that ANL integration was restricted to hippocampal neurons in RC3xCamk2a-Cre mice. Scale bar = 40 μm. (**E**) BONCAT-WB analysis of RC3xCamk2a-Cre mice reveals maximal ANL labelling when mice were administered 100 μg ANL/gbw via i.p. injection (n = 3 mice, one way ANOVA, Tukey's multiple comparison test, *p≤0.05). (**F**) BONCAT-WB analysis of RC3xCamk2a-Cre mice reveals maximal ANL labelling approximately 16 hr post injection (n = 3 mice, one way ANOVA, Tukey's multiple comparison test **p≤0.01, ***p≤0.001, ****p≤0.0001).

similarly to PBS-treated controls in a probe trial conducted 16 hr post-training, demonstrating that AHA treatment does not interfere with spatial LTM formation (*Figure 1—figure supplement 1*).

To differentiate spatial LTM-induced protein synthesis from background levels, we compared mice trained in the 30 min APA ('trained mice') to 'non-trained' controls (*Figure 1D*). Each non-trained mouse underwent a yoked version of the 30 min APA protocol, where instead of being shocked upon entry into the designated shock zone, the animal received shocks at the same time as a trained mouse with which it was paired (*Figure 1D*). These control mice were therefore unable to

undergo spatial learning, while still being exposed to the same environment, number of foot shocks, and a similar level of stress. The latter was confirmed by analysing plasma levels of the stress-related hormone, corticosterone (*Gong et al., 2015*), which were found to be similar between trained and non-trained mice (*Figure 1—figure supplement 2*).

Immediately following the behavioural task, both trained and non-trained mice were administered 50 µg AHA/gbw intraperitoneally and perfused 16 hr later without undergoing a probe test (*Figure 1D*). Using FUNCAT to visualise AHA-labelled proteins, we observed an increase in this signal in the hippocampus of trained mice compared to non-trained controls (*Figure 1E*), indicating that protein synthesis is increased in this brain region during the formation of spatial LTM. Our findings, in combination with the wealth of data demonstrating that hippocampal protein synthesis is essential for spatial LTM formation and storage (*Jarome and Helmstetter, 2014*; *Kleinknecht et al., 2012*; *Poucet et al., 2003*), led us to examine in more detail how spatial LTM formation alters the de novo proteome in the hippocampus.

## Novel MetRS mutant transgenic mice enable cell-type-specific in vivo labelling of newly synthesised proteins

Following our observation that hippocampal protein synthesis is increased during spatial LTM formation, we sought to refine our approach by developing an experimental system for cell-specific NCAA labelling in order to examine protein synthesis specifically in neurons of the hippocampus. To restrict protein labelling to hippocampal neurons, we exploited the fact that ANL can only be incorporated into nascent proteins in cells expressing mutant tRNA synthetases, such as NLL-MetRS (*Figure 2A*). We therefore generated a mouse strain that expresses NLL-MetRS in a Cre-dependent manner, allowing tissue- or cell-type-specific incorporation of ANL. This mouse strain, referred to as ROSA26a Cre-inducible Click Chemistry (RC3) strain, was generated by inserting the RC3 transgene (consisting of NLL-MetRS fused to EGFP downstream of a floxed stop cassette) into the permissive ROSA26a locus (*Bouabe and Okkenhaug, 2013*) using CRISPR/CAS9-mediated genome editing (*Figure 2B*). Thus, upon expression of Cre-recombinase in mice, the floxed-STOP cassette is excised, enabling expression of NLL-MetRS-EGFP and incorporation of ANL into newly synthesised proteins (*Figure 2B*).

For our study, we crossed the RC3 mice with the Camk2a-Cre (T29-1) strain that constitutively expresses Cre recombinase specifically in hippocampal neurons (*Tsien et al., 1996*). We validated confinement of NLL-MetRS-EGFP expression to hippocampal neurons in the resulting double transgenic strain by immunohistochemistry (*Figure 2C*), and, following ANL injection, revealed a similarly confined labelling of de novo synthesised proteins (*Figure 2D*). We next used BONCAT followed by western blotting (BONCAT-WB) to examine the dynamics of ANL labelling in the hippocampus. This allowed us to confirm that ANL labelling followed a similar time course to that of AHA labelling, with a dosage of 100 µg ANL/gbw resulting in maximal labelling at 16 hr (*Figure 2E&F*). We also confirmed that this labelling did not prevent spatial LTM formation, with ANL-treated RC3xCamk2a-Cre mice being able to recall the location of the shock-zone a week after training (*Figure 3—figure supplement 1*).

## De novo proteomic analysis identifies altered hippocampal synthesis of select proteins and pathways during spatial LTM formation

We next sought to identify de novo proteomic changes which occur in the hippocampal neurons during spatial LTM consolidation. We therefore induced spatial LTM formation in 5 month-old female RC3xCamk2a-Cre mice using the 30 min APA protocol (*Figure 3B*). This was immediately followed by intraperitoneal injection of 100 µg ANL/gbw of both trained and non-trained mice, which were perfused 16 hr later (*Figure 3A*). Using FUNCAT and immunohistochemistry, we observed an increase in total protein synthesis in the hippocampal neurons of the trained mice compared to non-trained controls (*Figure 3C*), recapitulating the above AHA findings. Moreover, we confirmed that blocking protein synthesis through the administration of anisomycin prevented spatial LTM formation (*Figure 3—figure supplement 2*).

Next, we identified which proteins were altered in synthesis in hippocampal neurons during spatial LTM formation using mass spectrometry-based proteomics. For this, ANL-labelled proteins from trained and non-trained mice were purified using BONCAT, and analysed via sequential window

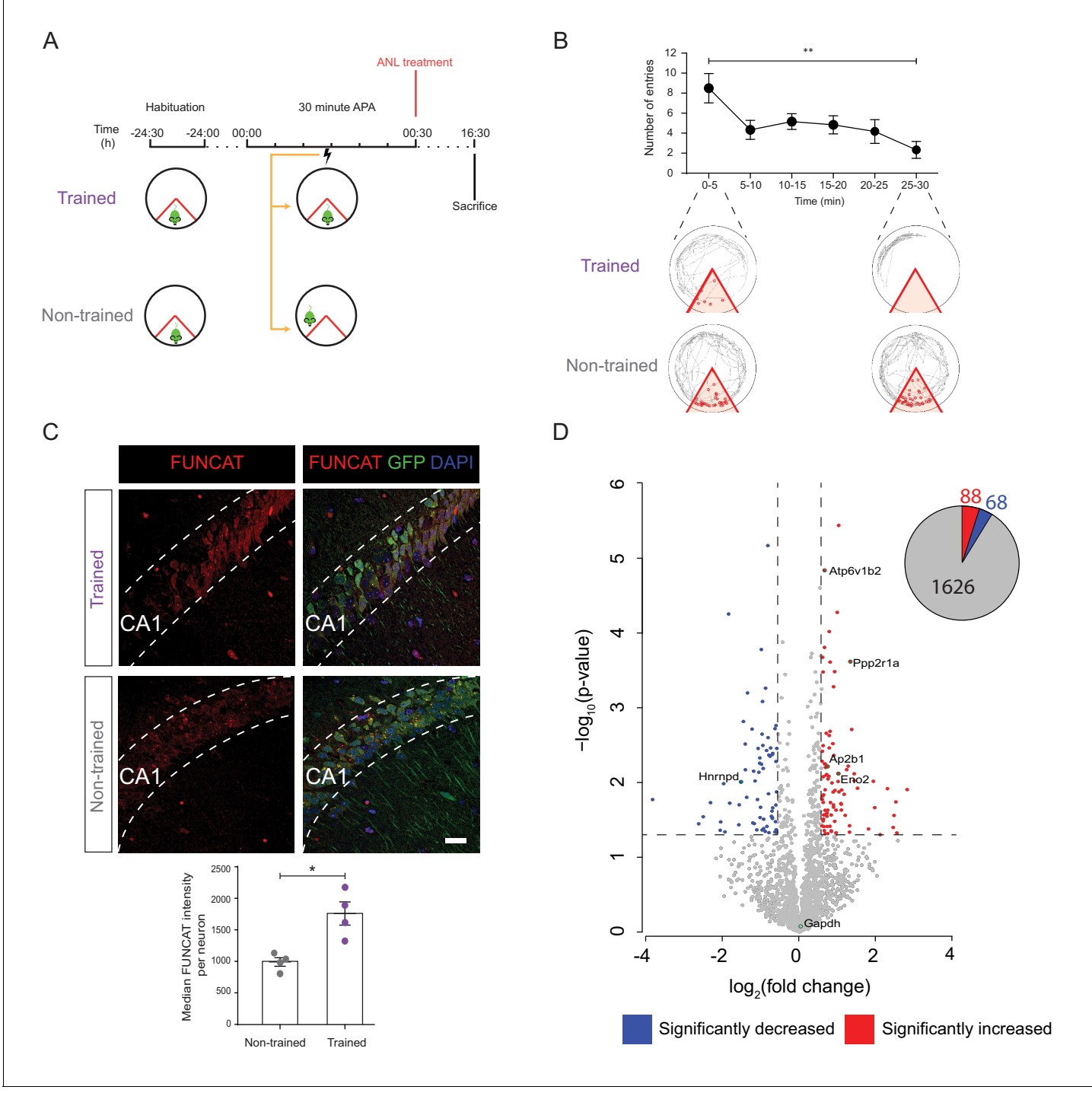

**Figure 3.** De novo proteomic analysis reveals altered hippocampal synthesis of selected proteins during spatial LTM formation. (**A**) Schematic of the 30 min APA task for trained and non-trained (yoked) RC3xCamk2-Cre mice. Trained mice received foot shocks upon entry into the designated shock zone, while non-trained mice were paired by receiving foot shocks at the same time as their trained partner. Upon completion of the 30 min APA task, mice were administered ANL and then perfused 16 hr later. (**B**) RC3xCamk2a-Cre mice trained in the 30 min APA task reduced the number of entries (red circles) into the shock zone (red) over the 30 min training period (n = 4 mice, one-way ANOVA, Dunnett's MCT, **p≤0.01). (**C**) Following spatial LTM formation, total protein synthesis was significantly increased in the hippocampal neurons of RC3xCamk2a-Cre mice. This was reflected by the increased FUNCAT signal observed in trained compared to non-trained mice (n = 4 mice, 30 neurons per mouse, Student's paired t-test, *p≤0.05). Scale bar = 40 μm. (**D**) Volcano plot representing the relative abundance of de novo synthesised proteins in the hippocampus of trained and non-trained RC3xCam2ka-Cre mice. In total, 1782 proteins were quantified in four trained and non-trained mice each, using BONCAT-SWATH-MS. Proteins which were significantly increased in synthesised in trained mice (fold-change ≥1.5, p≤0.05) are shown in red, whereas proteins which exhibited significantly

*Figure 3 continued on next page*

*Figure 3 continued*

decreased synthesis (fold-change ≤0.66, p≤0.05) are shown in blue (n = 4 mice, Student's t-test). Subsequently validated proteins are encircled in green (see *Figure 5*).

The online version of this article includes the following source data and figure supplement(s) for figure 3:

**Source data 1.** ANL treated 30 minute APA behavioural data for protemic analysis.
**Figure supplement 1.** RC3xCamk2a-Cre mice treated with ANL and trained in the 30 min APA task were able to recall the location of the shock zone a week after training.
**Figure supplement 1—source data 1.** ANL treated 30 minute APA behavioural data.
**Figure supplement 2.** The spatial long-term memory formation observed in the 30 min APA task is dependent upon protein synthesis.
**Figure supplement 2—source data 1.** Anisomycin treated 30 minute APA behavioural data.
**Figure supplement 3.** Proteomic analysis of Anl labelled proteins from trained and non-trained RC3xCamk2a-Cre mice.

acquisition of all theoretical fragment ions mass spectrometry (SWATH-MS). Of the 1782 proteins quantified, 156 were identified to have significantly altered synthesis ($|FC| \geq 1.5$, p≤0.05) in trained mice compared to non-trained controls, comprising of 88 proteins showing increased and 68 proteins showing decreased synthesis (*Figure 3D*). Euclidian clustering analysis revealed that the de novo proteomes of trained mice were more similar to each other than those of their paired non-trained controls, demonstrating a distinctive change in the proteome composition of the experimental groups (*Figure 3—figure supplement 3B*).

The 156 proteins with significantly altered synthesis during spatial LTM formation were next analysed using STRING network analysis. Of these proteins, 125 ($\approx$80%) formed a highly interconnected, large network, with a median of four protein-protein interactions per node (*Figure 4*). Using the Molecular Complex Detection (MCODE) clustering algorithm (*Bader and Hogue, 2003*), we identified five distinct clusters of de novo synthesised proteins (*Figure 4*). Gene ontology (GO), KEGG and Reactome analysis revealed that the proteins in these clusters were associated with mRNA splicing, ATP hydrolysis coupled proton transport, vesicle-mediated transport, biogenesis of complex I, and Rho GTPase effectors (*Figure 4*).

## Validation of key proteins identified in the de novo proteomic analysis using BONCAT western blotting

Given that our de novo proteomic analysis revealed distinct changes in hippocampal protein synthesis during LTM formation, we next performed a validation of a subset of proteins within our identified clusters using BONCAT-WB. This revealed increased synthesis of α-adaptin (*Ap2a1*), neuron specific enolase (NSE: *Eno2*), V-ATPase subunit B2 (V-ATPase B2: *Atp6v1b2*), and the α isoform of the structural subunit A of protein phosphatase 2A (PP2A-A: *Ppp2r1a*), in the hippocampal neurons of trained mice (*Figure 5*). We also confirmed that the synthesis of ARE/poly(U)-binding/degradation factor 1 (AUF-1: *Hnrnpd*) was decreased 16 hr following training (*Figure 5*). Lastly, as an unaltered control, we examined the synthesis of the housekeeping protein glyceraldehyde 3-phosphate dehydrogenase (GAPDH). As expected from our BONCAT-SWATH-MS analysis, BONCAT-WB revealed that its synthesis was not altered during spatial LTM formation (*Figure 5*). Together, these results validate our de novo proteomic analysis and provide further evidence that the formation of spatial LTM is associated with regulated changes in the synthesis of specific proteins in hippocampal neurons.

## Discussion

In our study, we used NCAA labelling in combination with FUNCAT, BONCAT and SWATH quantitative proteomics to examine how de novo protein synthesis is altered in the initial stages of spatial LTM consolidation. Using FUNCAT labelling with the non-canonical amino acid AHA, we first observed that as expected, protein synthesis was increased in the hippocampus of mice during spatial LTM formation (*Jarome and Helmstetter, 2014*). We further refined our approach by generating the RC3 mouse strain, which, upon crossing with the Camk2a-Cre strain and administration of ANL, allowed for neuron-specific labelling of de novo synthesised proteins specifically in the hippocampus. Using BONCAT in combination with SWATH-MS quantitative de novo proteomics, we identified a total of 1782 proteins which were newly synthesised in hippocampal neurons, 156 of which were

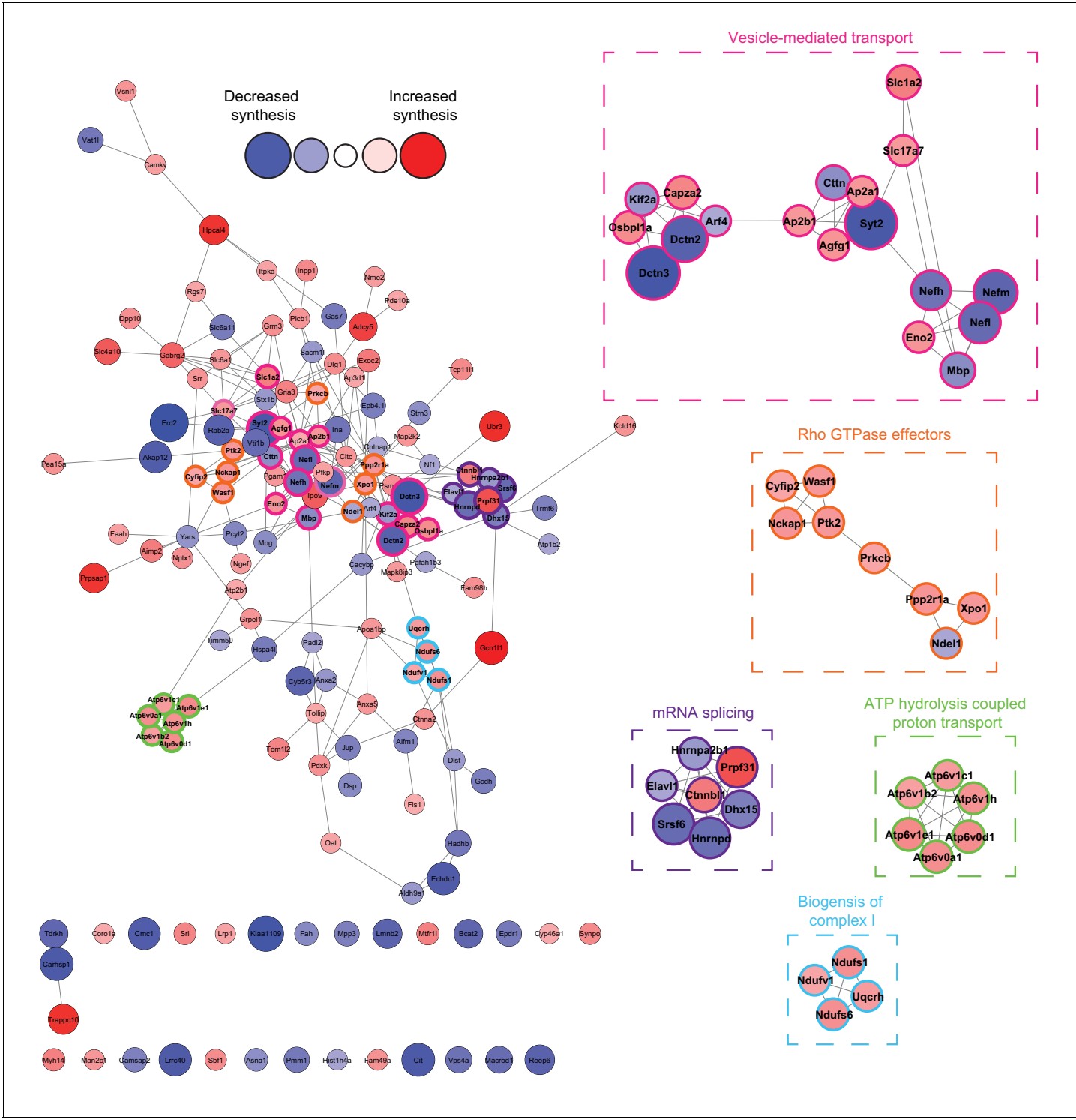

**Figure 4.** Network analysis reveals that the synthesis of distinct clusters of proteins is significantly altered during spatial LTM formation. Network analysis using the STRING database reveals that of the 156 proteins identified to be altered in synthesis during spatial LTM formation (|fold-change| ≥ 1.5, p≤0.05), 125 (≈80%) showed evidence of interaction (STRING score cut off ≥0.4) with at least one other significantly regulated protein, forming a highly interconnected network. Within this network, there was a median of 4 interactions per node. MCODE cluster analysis revealed the presence of 5 distinct clusters which were associated with mRNA splicing, ATP hydrolysis-coupled proton transport, vesicle-mediated transport, biogenesis of mitochondrial complex I, and Rho GTPase effectors. Proteins in clusters are depicted by a coloured border and are magnified in the inserts. The distance between each node is representative of the STRING score. Proteins which did not display interactions are not shown. The absolute fold-change is represented by the node size, and the directionality of the fold-change by the node colour.

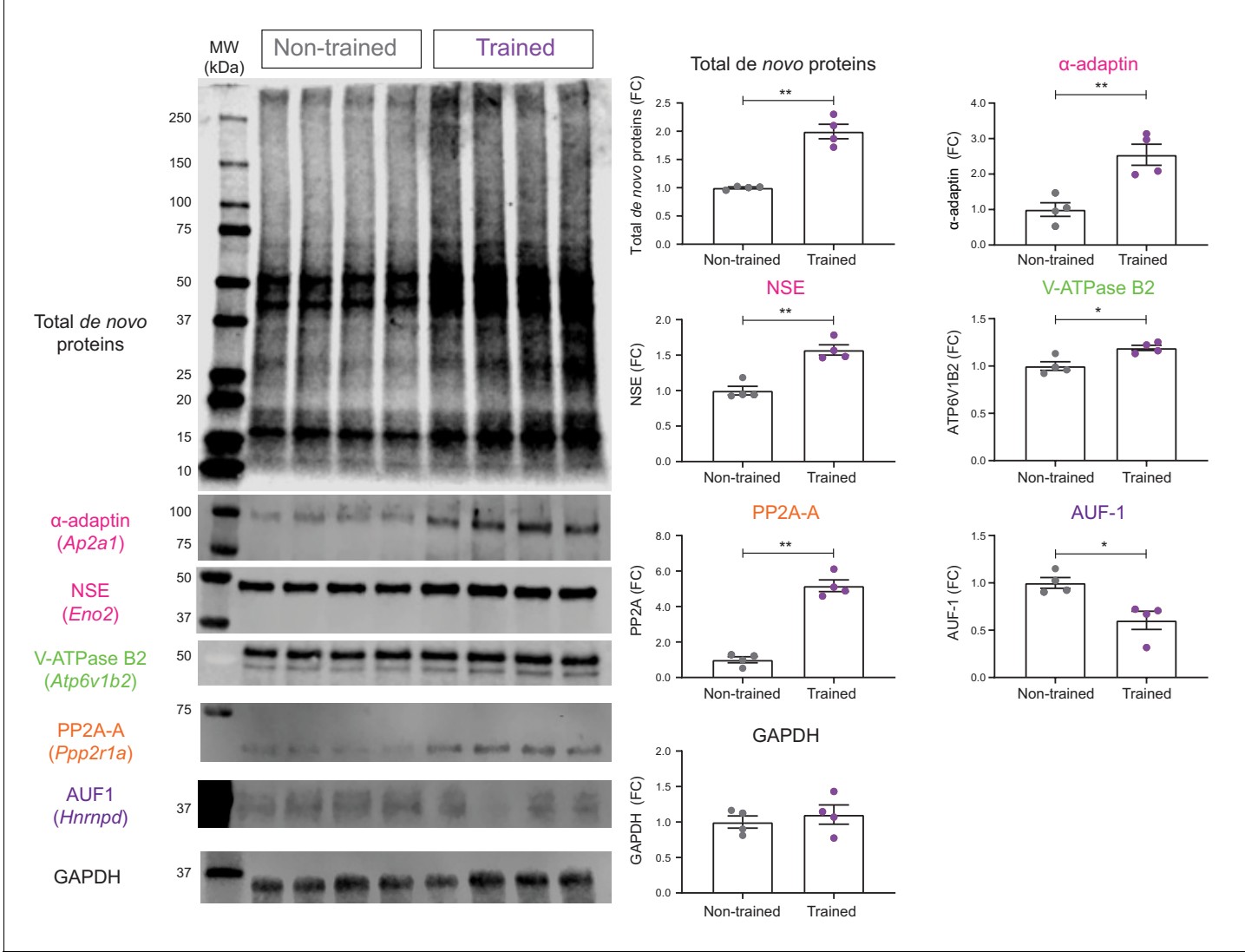

**Figure 5.** Validation of de novo proteomic comparison of trained vs non-trained. RC3xCamk2a-Cre mice using BONCAT western blotting BONCAT-WB confirms that in trained mice, there is increased hippocampal synthesis of α- α-adaptin (*Ap2a1*), neuron specific enolase (NSE: *Eno2*), V-ATPase subunit B2 (V-ATPase B2: *Atp6v1b2*), and the α isoform of the structural subunit of protein phosphatase 2A (PP2A-A: *Ppp2r1a*) compared to non-trained controls. Synthesis of the ARE binding protein ARE/poly(U)-binding/degradation factor 1 (AUF-1: *Hnrnpd*) was also confirmed to be decreased during spatial LTM formation. The synthesis of the housekeeping gene, glyceraldehyde 3-phosphate dehydrogenase (GAPDH), was unchanged in the hippocampal neurons of trained RC3xCamk2a-Cre mice compared to non-trained controls (p=0.48), together validating the SWATH analysis (n = 4 mice, Student's paired t-test, *p≤0.05, **p≤0.01, ***p≤0.001).

The online version of this article includes the following figure supplement(s) for figure 5:

**Figure supplement 1.** Complete images of the western blots shown in *Figure 5*.

found to be significantly altered in synthesis following spatial LTM formation, with BONCAT-WB being used to independently validate a subset of these proteins. These regulated proteins are involved in a wide range of cellular functions, including mRNA splicing, ATP hydrolysis-coupled proton transport, vesicle-mediated transport, biogenesis of mitochondrial complex I, and signalling through Rho GTPases. Our de novo proteomic data suggest a role for these proteins, and the pathways they are involved in, in spatial LTM formation and, more generally, demonstrate that NCAA labelling is a robust, non-biased experimental strategy to examine how the de novo proteome is altered during complex rodent behaviour.

Previously, NCAA labelling has been used in rodents to observe changes in de novo protein synthesis during development (*Calve et al., 2016*), to compare different cell-types following

environmental enrichment (*Alvarez-Castelao et al., 2017*), and to examine changes which occur in mouse models of disease (*Evans et al., 2019*; *McClatchy et al., 2015*). To the best of our knowledge, the current study represents the first use of NCAA labelling to examine spatial LTM formation in mice.

NCAAs, such as AHA and ANL, have typically been delivered to mice through diet, with labelling periods ranging from 6 to 21 days (*Alvarez-Castelao et al., 2017*; *McClatchy et al., 2015*). Spatial LTM consolidation is dependent on multiple temporally spaced phases of protein synthesis (*Fioriti et al., 2015*; *Meiri and Rosenblum, 1998*; *Ozawa et al., 2017*), with the first 24 hr following training being the most sensitive to the inhibition of protein synthesis (*Freeman et al., 1995*; *Quevedo et al., 1999*). Given that the majority of changes to the hippocampal de novo proteome are thought to occur within this time frame, we considered the longer labelling periods used in previous studies not to be suitable for examining spatial LTM-induced protein synthesis. We therefore delivered AHA and ANL via intraperitoneal injection rather than dietary administration, enabling us to label newly synthesised proteins within a 16 hr time window (*Figure 2F*). An additional advantage of labelling for this shorter period is the increased temporal specificity, with a higher proportion of tagged proteins being synthesised only in response to the behavioural paradigm studied, compared to those that would be synthesised after a longer period of time.

Using ANL labelling in combination with SWATH-MS, we obtained a similar degree of proteome coverage to the only other study we are aware of that has used ANL labelling in mice (*Alvarez-Castelao et al., 2017*). However, given that the neuronal proteome is believed to consist of around 6000 proteins (*Schanzenbächer et al., 2016*), more invasive delivery techniques such as cranially implanted osmotic pumps may be required to achieve a deeper degree of labelling than observed in either study.

In order to investigate protein synthesis specifically during spatial LTM formation, we established an accelerated version of the five-day APA paradigm, which relies upon a singular training event (*Figure 1C*), making it easier to capture the time-period during which spatial LTM formation occurs. We confirmed that this 30 min APA task induces spatial LTM by examining two defining characteristics of long-term memory; the ability to persist for long periods of time and the requirement of new protein synthesis (*Figure 3—figure supplement 1*, *Figure 3—figure supplement 2*) (*Costa-Mattioli et al., 2009*; *Rosenberg et al., 2014*).

Mice that are exposed to a novel environments and stimuli could potentially experience increased levels of stress while undertaking behavioural tasks, which may alter protein synthesis (*Chandran et al., 2013*; *Moncada and Viola, 2007*). We controlled for this by using paired, yoked controls and confirmed that these non-trained mice showed similar levels of stress as their trained counterparts (*Figure 1—figure supplement 2*). We therefore conclude that the observed changes in protein synthesis between trained and non-trained mice were likely induced by spatial LTM formation, rather than confounding factors including training and stress.

In our analysis we focused on the hippocampus, as protein synthesis in this region plays a critical role in spatial LTM (*Burgess et al., 2002*; *Jarome and Helmstetter, 2014*; *Kleinknecht et al., 2012*; *Merlo et al., 2015*). As our initial AHA labelling and FUNCAT analysis confirmed that protein synthesis is altered in the hippocampus during spatial LTM formation (*Figure 1E*), we chose to use cell-type-specific NCAA labelling in order to restrict de novo protein labelling to hippocampal neurons in vivo.

To restrict the expression of NLL-MetRS to hippocampal neurons, we crossed RC3 mice with the Camk2a-Cre T29-1Stl/J mouse strain. When originally generated, this mouse strain expressed Cre-recombinase only in the CA1 pyramidal neurons of the hippocampus and the testes (*Tsien et al., 1996*); however, more recent analyses revealed expression in other regions of the hippocampus (*Alvarez-Castelao et al., 2017*; *McGill et al., 2017*), possibly reflecting a genetic drift. Our results are consistent with these more recent studies, with RC3xCamk2a-Cre mice expressing NLL-MetRS-EGFP in the neurons of the CA1, CA2, CA3, as well as dentate gyrus regions of the hippocampus (*Figure 2D*).

Recently, another Cre-dependent mouse strain was established that expresses the mutant tRNA synthetase MetRS-L247G (*Alvarez-Castelao et al., 2017*). Unlike NLL-MetRS, which is considered to label proteins with ANL only at their amino-terminus (*Ngo et al., 2013*), MetRS-L247G also replaces internal methionine residues with ANL (*Müller et al., 2015*; *Yang et al., 2018*), which likely results in a wider range of protein labelling.

By combining hippocampal neuron-specific ANL labelling with BONCAT-SWATH-MS, we identified 156 proteins which were significantly altered in synthesis during spatial LTM formation induced by the 30 min APA paradigm (*Figure 3D*). While these changes represent a small proportion of the total-ANL proteome, these findings are in line with previous studies assessing the total proteome in different stages of memory consolidation (*Borovok et al., 2016*). STRING analysis of the 156 significantly altered proteins revealed that the vast majority of these proteins interact with a least one other significantly altered protein, forming a large, highly interconnected network (*Figure 4*). This high degree of interconnection would suggest that, as expected, spatial LTM formation induces changes not only in the synthesis of specific proteins but also pathways.

Interestingly, while FUNCAT analysis revealed an overall increase in the amount of new protein synthesis during spatial LTM formation (*Figure 3B*), our de novo proteomic analysis identified a similar number of proteins with decreased and increased synthesis (68 and 88, respectively) in the 16 hr following training. This would suggest that rather than simply inducing the synthesis of memory-related proteins, spatial LTM memory formation causes neurons to selectively modulate certain pathways and molecular processes.

MCODE analysis of the network-regulated proteins identified five distinct clusters of newly synthesised proteins. These were associated with mRNA splicing, ATP hydrolysis-coupled proton transport, vesicle-mediated transport, biogenesis of mitochondrial complex I, and Rho GTPase effectors. We confirmed the validity of our de novo analysis by examining a number of proteins from these clusters using BONCAT-WB, with all of the examined proteins showing similar changes in synthesis for both detection methods (*Figure 5*).

Many of the proteins and pathways identified by our analysis have previously been associated with memory formation. Examples are α- and β-adaptin (*Ap2b1*), which were increased in synthesis during spatial LTM formation (*Figure 4* and *Figure 5*). These two proteins are subunits of the clathrin adaptor protein 2 (AP2) complex, which regulates synaptic levels of GluR1- and GluR2/GluR3-containing AMPA receptors (*Garafalo et al., 2015*). This process is important in the memory-related processes of long-term potentiation (LTP) and long-term depression (LTD) (*Barth and Wheeler, 2008*; *Hardt et al., 2014*), as well as complex rodents behaviours such as visual recognition memory, which is disrupted when the interaction between AP2 and the GluR2 subunit is blocked (*Griffiths et al., 2008*).

Another memory-related protein identified to be altered in synthesis in our de novo proteomic analysis was the structural subunit A of protein phosphatase 2A, PP2A-A. Both BONCAT-SWATH-MS and BONCAT-WB found that the α isoform of PP2A-A was increased in synthesis during the 16 hr following training (*Figure 3D* and *Figure 5*). Inhibition of PP2A has been demonstrated to block LTP (*Belmeguenai, 2005*), with hippocampal PP2A knock-out animals presenting altered extinction of long-term memory (*Wang et al., 2019*).

Our de novo proteomic analysis also identified several molecular processes which have yet to be robustly linked to spatial LTM formation. One such process is the regulation of mRNA splicing. We observed that during memory and learning, there was increased synthesis of pre-mRNA-processing factor 31 (Prp31) and β-catenin-like protein 1 (Ctnnbl1), both of which are required for the formation and activation of the spliceosome (*Ganesh et al., 2011*; *Yuan et al., 2005*) (*Figure 4*), while the synthesis of the pre-mRNA-splicing factor ATP-dependent RNA helicase DHX15 (DHX15), which is involved in the disassembly of spliceosomes, was decreased (*Wen et al., 2008*) (*Figure 4*). We further observed decreased synthesis of serine/arginine-rich splicing factor 6 (SRSF6), which inhibits the splicing of exon 10 of the microtubule associated protein tau (*Yin et al., 2012*). Taken together, these results suggest that during spatial LTM formation, there may be an alteration the activity of spliceosomes, leading to altered splicing of certain mRNAs such as *MAPT,* although further examination will be required to confirm if these molecular process are altered in spatial LTM.

The current working model of spatial LTM formation suggests that following spatial training, hippocampal neurons undergo protein synthesis in response to certain stimuli (*Kandel et al., 2014*; *Squire et al., 2015*). In our study, we used cell-type-specific NCAA labelling, in combination with a novel 30 min APA protocol and quantitative SWATH-MS de novo proteomics, to examine how the hippocampal de novo proteome is changed during spatial LTM formation. We found that in hippocampal neurons, there was altered synthesis of specific sets of proteins associated with a diverse, yet interconnected set of neuronal pathways and cellular processes following spatial training. In addition to observing altered synthesis of a number proteins already associated with memory, such as α-

adaptin, β-adaptin and PP2A-A, we also identified alterations in mRNA splicing as a potential neuronal mechanism which underpins spatial LTM formation. More generally, our findings highlight the potential of cell-type specific NCAA labelling using transgenic mouse strains such as the RC3 mice as a robust tool for identifying and characterising cell-type-specific changes in de novo protein synthesis that occur in response to a wide range of both physiological and pathological stimuli, including complex rodent behaviours.

## Materials and methods

### Animals and ethics
4–5 month old female C57BL/6, ROSA26a Cre-inducible Click chemistry (RC3), and RC3 mice crossed with Camk2a-Cre T29-1 Stl/j mice (Jackson Labs, 0005359) were used. Mice were provided access to food and water and housed on a 12 hr light/dark cycle. All experiments were approved by and carried out in accordance with the guidelines of the Animal Ethics Committee of the University of Queensland (QBI/554/17/NHMRC).

### Generation of RC3 mouse strain
The RC3 donor plasmid was obtained by subcloning the NLL-MetRS-EGFP transgene into the ROSA26A-targetting mammalian expression vector Ai2 (Ai2 was a gift from Hongkui Zeng, Addgene plasmid #22796). The RC3 transgenic mouse strain was generated by CRISPR/CAS9-mediated insertion of the linearized RC3 donor into the ROSA26 locus using a previously published sgRNA (*Chu et al., 2015*). The CAS9-sgRNA complexes and linearized donor plasmid were introduced into fertilised eggs by pronuclear injection as previously described, with minor modifications (*Ittner and Götz, 2007*). Offspring were genotyped by PCR using EGFP genotyping primers, with positive pups being confirmed by Sanger sequencing conducted at the Australian Equine Genetics Research Centre (AEGRC).

### Non-canonical amino acid treatment of mice
The non-canonical amino acid AHA (ThermoFisher, C10102) was dissolved in phosphate-buffered saline (PBS) and administered to wild-type mice as previously described (*Evans et al., 2019*). Optimal ANL (Jena Bioscience, CLK-AA009) labelling conditions were examined using BONCAT-WB. A dosage of 100 μg Anl/gbw and a labelling period of 16 hr resulted in maximal labelling and was used for all further experiments.

After treatment, mice were deeply anaesthetised with pentobarbitone sodium and then intracardially perfused with 25 mL of PBS. The brains were subsequently dissected, with one hemisphere being processed for immunohistochemistry and FUNCAT, and the other for BONCAT.

### Behavioural analysis
Spatial memory was assessed using a modified APA test. In brief, mice were trained over 30 min to use spatial cues to avoid a shock zone within an arena rotating at 1 rpm. Mice were handled daily for 2 min over a seven day period, prior to being habituated to the rotating arena during a 30 min exploration session. 24 hr after habituation, mice were placed into the arena with a fixed 60° shock zone extending from the centre point of the arena to the southern side of the room.

Mice were separated into two groups, trained and non-trained. Trained mice received a 0.5 mA shock upon entry into the shock zone, with an entrance shock delay of 0.5 s, and a 1.5 s interval between shocks. The number of shocks received and the number of entries into the shock zone were analysed in 5 min intervals. In the non-trained group, mice were paired to individual littermates from the trained group, receiving shocks at the same time as the trained mouse, irrespective of the spatial location of the non-trained mouse. Behavioural analysis was performed at the same time on sequential days in order to control from differences in protein synthesis due to circadian rhythm. Following sacrifice, further sample preparation and analysis for trained and non-trained mice was performed simultaneously.

In order to determine if the mice trained with the 30 min APA underwent spatial LTM formation, they were assessed for their ability to recall the location of the shock zone in 5 min probe trials, where mice did not receive shocks, conducted either 16 hr, 24 hr or 1 week after training. Probe

trials were also used to confirm that NCAA treatment did not interfere with spatial LTM formation. In experiments where protein synthesis was inhibited, mice were administered anisomycin (Sigma-Aldrich, A9789) 150 µg/gwb via subcutaneous injection immediately after training as this has been previously demonstrated to inhibit hippocampal protein synthesis for >9 hr (*Wanisch and Wotjak, 2008*). For all trained and non-trained experiments, mice were administered 50 µg AHA/gwb or 100 µg ANL/gwb immediately after training and were perfused 16 hr later without undergoing a probe trial.

## Plasma corticosterone quantification

In order to assess stress levels in mice during the 30 min APA, trained, non-trained and naïve (habituated but not shocked) mice were sacrificed immediately following their respective behaviour tasks. Blood was then collected via cardiac puncture and left to clot for 1 hr, with plasma then being collected via centrifugation. Plasma corticosterone levels were quantified in three mice from each group via ELISA which was performed in triplicate (Enzo Life Sciences, ADI-900–097).

## FUNCAT and immunohistochemical analysis

Following PBS perfusion, brain hemispheres were fixed in 4% paraformaldehyde for 24 hr and then placed in cryoprotectant solution (30% glycerol, 30% ethylene glycol in 1x PBS) for 48 hr at 4°C. 25 µm thick free floating sections were then cut between Bregma −1.34 and −2.06 µm using a vibratome (Leica VT1000).

Prior to FUNCAT staining sections were placed in blocking solution (1% bovine serum albumin (BSA), 0.05% Tween in PBS) for 1 hr at room temperature, with three sections per mouse being analysed. AHA- and ANL- labelled proteins were then visualised by incubating sections with 6.25 µM Alexa555-DIBO (ThermoFisher, C20021) in blocking solution overnight at 4°C under constant agitation. Neurons were visualised by staining with a MAP2 antibody (Abcam, ab5392, 1:1000) and anti-chicken Alexa Fluor647 (ThermoFisher, A21449, 1:1000). Sections were washed repeatedly with 0.05% Tween in PBS and then stained with DAPI. As negative controls, sections of PBS-treated mice stained with Alexa555-DIBO were used throughout all experiments. Images were taken using a Zeiss 710 laser scanning confocal microscope.

## BONCAT purification

Following PBS perfusion the hippocampus of RC3xCamk2a-Cre mice was dissected. The samples were snap-frozen and then extracted in radioimmunoprecipitation assay (RIPA) buffer (Cell Signalling, 9806) as previously described (*Bodea et al., 2017*), with protein concentrations being determined using the bicinchoninic acid (BCA) assay (ThermoFisher, 23225).

BONCAT purification was carried out as previously described (*Evans et al., 2019*). Briefly, for samples to be analysed via western blot following BONCAT purification (BONCAT-WB), 100 µg of protein was used. Proteins were first alkylated with iodoacetamide (IAA) as this has been shown to reduce non-specific reactions when using the strain-promoted azide-alkyne cycloaddition (*van Geel et al., 2012*). Anl labelled proteins were then reacted with 100 µM DIBO-biotin (Click Chemistry Tools, A112) for 2 hr at room temperature. 40 µg of streptavidin-coated Dynabeads (ThermoFisher, 11205D) were then used to purify biotinylated proteins, with beads being washed multiple times with IP wash buffer (0.1% SDS and 0.05% Tween in Tris-buffered saline (TBS)). Bound proteins were removed from the beads by boiling in 1x Laemmli buffer.

For BONCAT-SWATH-MS analysis, samples were purified as above, but using 250 µg of protein and 100 µg of streptavidin-coated Dynabeads per sample. Beads were then washed in IP wash buffer and resuspended in TBS.

## Western blot analysis

Following BONCAT purification, equal volumes of the elution fraction were loaded and separate by SDS-PAGE and analysed via western blotting as previously described (*Evans et al., 2019*). The total amount of newly synthesised proteins was quantified using the REVERT total protein stain (LI-COR, 926–11010), with representative proteins for each cluster were detected using the following primary antibodies: α adaptin (ThermoFisher, MA3-061, 1:500), neuron specific enolase (NSE) (Abcam,

ab53025, 1:500), AT6V1B2 (Abcam, ab73404, 1:500), PP2A-A (Sigma-Aldrich, 07–250, 1:1000), AUF-1 (Sigma-Aldrich, 07–260, 1:500), and GAPDH (Millipore, MAB374, 1:1000).

## BONCAT-SWATH-MS analysis

In order to identify proteins newly synthesised during spatial memory formation, Anl-labelled proteins from both trained and non-trained RC3xCamk2a-Cre mice were analysed by BONCAT-SWATH-MS mostly as previously described (*Evans et al., 2019*). Briefly, BONCAT-purified proteins bound to beads from four trained and four untrained samples were placed in Triethylammonium bicarbonate (TEAB) buffer and subsequently reduced with DTT, followed by alkylation with iodoacetamide. Samples were then digested with 80 ng of trypsin overnight. For generation of the custom ion library, samples were then pulled and resuspended in 5 mM ammonium hydroxide solution (pH 10.5). Peptides were then fractionated using high pH RP-HPLC, with the resulting 17 fractions being analysed via non-LC MS/MS to form the custom ion library used for SWATH analysis. In order to control for background purified proteins, BONCAT purification was performed on an RC3xCamk2a-Cre PBS-treated negative control. Peptides identified from this negative control were excluded from further analysis, in addition to peptides identified in previous BONCAT-SWATH-MS negative control experiments (*Figure 3—figure supplement 3A*, *Supplementary file 1*).

Using this custom ion library, samples were then analysed using SWATH-MS using a false discovery rate (FDR) of 1% for peptide and protein identification as previously described (*Evans et al., 2019*).

## Bioinformatic analysis SWATH-MS data

Proteins identified using SWATH-MS were statistically compared using paired t-tests. We opted to use an unadjusted p-value cut off of $p \leq 0.05$ and an absolute fold-change cut off of $\geq 1.5$ to identify proteins with significantly altered synthesis. This is because we observed that correcting for multiple comparisons greatly reduced the number of significantly altered proteins from 156 to 27, resulting in the exclusion of a number of true positives, such as NSE and AUF-1, which were confirmed to be altered in synthesis using BONCAT-WB (*Figure 5*, *Supplementary file 2*). This culling effect on true positives is commonly observed in proteomics studies with smaller sample sizes using multiple comparisons (*Pascovici et al., 2016*). Previous SWATH-MS experiments using spiked lysates have demonstrated that the cut offs used in our study result in estimated quantitative FDRs of <10% (*Wu et al., 2016*). Similar cut offs were also used in other studies (*Ganief et al., 2017*; *Liu et al., 2019*). Network analysis was performed using Cytoscape (v3.6.0). Data from the SWATH-MS analysis were mapped to the STRING protein query database for *Mus musculus* using the UniProt identifier. A confidence of interaction score cut-off of 0.4 was used. A network map of the 156 proteins which exhibited significantly altered synthesis in trained mice was then generated using the edge-weighted spring-embedded layout. Clusters of regulated proteins were identified using Molecular Complex Detection (MCODE). The proteins in these clusters where then analysed using the GO, KEGG and Reactome databases. Cluster names, which were both informative and contained a majority of proteins within the cluster, were manually assigned. Heatmapper was used to perform a Euclidian clustering analysis of the relative abundance of all 1782 quantified proteins for each sample (*Babicki et al., 2016*).

## Image analysis

Analysis of images obtained by confocal microscopy was performed blinded and carried out using ImageJ. Protein synthesis in AHA-treated mice was quantified by measuring the median FUNCAT intensity in rectangular regions of interest of the same size around the CA1 neurons of the hippocampus, with each data point representing the mean of three sections analysed per animal. For RC3xCamk2a-Cre mice, the presence of NLL-MetRS-EGFP expression enabled FUNCAT intensity to be quantified per neuron. ANL labelling was quantified by measuring the median FUNCAT intensity, with each data point representing the mean of 30 hippocampal CA1 neurons analysed per animal. Western blots were analysed using the LI-COR Light Studio software.

## Statistical analysis

Statistical analysis was performed in GraphPad Prism 7.0 software using Student's paired t-test, Student's unpaired t-test, one way ANOVA, or two-way ANOVA, with Tukey's multiple comparisons test (MCT), Sidak's MCT, or Dunnett's MCT being used as appropriate. All values are given as mean ± standard error of the mean (SEM).

## Data availability

The RAW mass spectrometry proteomics data used in this study has been deposited to the ProteomeXchange Consortium via the PRIDE partner repository (*Perez-Riverol et al., 2019*) with the dataset identifier PXD015820.

## Acknowledgements

The authors wish to thank Daniel Blackmore, Jessica Barbizzi, Tishila Palliyaguru, Linda Cumner and Trish Hitchcock for their excellent technical support along with Rowan Tweedale for critical reading of the manuscript. We also like to thank Xiaomin Song and the Australian Proteomic Analysis Facility for performing the SWATH-MS analysis. This research was supported by the Estate of Dr Clem Jones AO, the State Government of Queensland, the Federal Government of Australia (ACT900116), and by grants from the Australian Research Council (ARC DP160103812). LGB is supported by the Peter Hilton Fellowship.

## Additional information

### Funding

| Funder | Grant reference number | Author |
| --- | --- | --- |
| The Estate of Dr Clem Jones AO | | Jürgen Götz |
| Australian Research Council | ARC DP160103812 | Jürgen Götz |
| Queensland Government | ACT900116 | Jürgen Götz |
| The Peter Hilton Fellowship | | Liviu-Gabriel Bodea |
| National Health and Medical Research Council | GNT11457569 | Liviu-Gabriel Bodea Jürgen Götz |

The funders had no role in study design, data collection and interpretation, or the decision to submit the work for publication.

### Author contributions

Harrison Tudor Evans, Conceptualization, Investigation, Visualization, Methodology; Liviu-Gabriel Bodea, Jürgen Götz, Conceptualization, Supervision, Funding acquisition

### Author ORCIDs

Harrison Tudor Evans https://orcid.org/0000-0003-0322-0554
Jürgen Götz https://orcid.org/0000-0001-8501-7896

### Ethics

Animal experimentation: All experiments were approved by and carried out in accordance with the guidelines of the Animal Ethics Committee of the University of Queensland (QBI/554/17/NHMRC).

### Decision letter and Author response

Decision letter https://doi.org/10.7554/eLife.52990.sa1
Author response https://doi.org/10.7554/eLife.52990.sa2

## Additional files

### Supplementary files

• Supplementary file 1. List of peptides identified by 2D-LC MS/MS from BONCAT purified from ANL-treated RC3xCamk2a-Cre mice and PBS-treated negative controls.

• Supplementary file 2. SWATH-MS data from quantification of BONCAT purified proteins from trained and non-trained RC3xCamk2a-Cre. For each of the 1782 proteins quantified, the name, Uniprot identifier, gene name, fold change in trained compared to non-trained mice, p-value, Benjamini and Hochberg adjusted p-value, number of peptides sequences used for quantified, and the normalised protein peak area are given. Proteins contained within clusters identified by MCODE analysis are highlighted.

• Transparent reporting form

### Data availability

All data in this study is presented in the included manuscript, supporting files, and source data files. The proteomics data set has been uploaded to the PRIDE repository with the dataset identifier PXD015820.

The following dataset was generated:

| Author(s) | Year | Dataset title | Dataset URL | Database and Identifier |
|---|---|---|---|---|
| Evans HT, Bodea LG, Götz J | 2019 | Cell-specific non-canonical amino acid labelling identifies changes in the hippocampal de novo proteome during memory formation | https://www.ebi.ac.uk/pride/archive/projects/PXD015820 | PRIDE, PXD015820 |

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
