## [Decision Letter]

Thank you for submitting your work entitled "Cell-specific non-canonical amino acid labelling identifies changes in the de novo proteome during memory formation" for consideration by *eLife*. Your article has been reviewed by three peer reviewers, including Sacha B Nelson as the Reviewing Editor and Reviewer #1, and the evaluation has been overseen by a Senior Editor. The following individual involved in review of your submission has agreed to reveal their identity: Erin Schuman (Reviewer #3).

As you will see below, the reviewers identified a number of distinct and overlapping problems with the present manuscript that preclude its acceptance despite significant enthusiasm for the overall topic and several features of the study. Performing the required additional experiments and analyses is likely to require significant additional effort, but the reviewers were in agreement that several of these issues would need to be addressed to support the conclusions of the paper. It is *eLife* policy to reject manuscripts for which significant additional work is required. In this case, if key issues can be addressed, we would encourage a resubmission, and would endeavor to ensure that it is handled by the same reviewers, although we cannot guarantee this.

In an effort to make this process as transparent as possible, we are including the original reviews, but note that some of the issues were viewed as more critical than others. Specifically:

1) All reviewers agreed that identifying background proteins associated with the purification was a critical control (e.g. Rev 3 point 3).

2) In addition, showing that the task requires protein synthesis was felt to be important.

3) Improvement of the statistical analysis was also felt to be critical.

4) Finally, clarifying what was learned about the underlying biology was also deemed important.

In contrast, after discussion, reviewers agreed that characterization of the specificity of the driver mouse could potentially be addressed by referring to other studies. It was also felt that the small number of proteins identified could be discussed without requiring additional experiments. Finally, the concern of reviewer 1 about the possibility of stress-dependent effects rather than learning being required for the changes seen could be addressed by the purity controls and demonstration that the task requires protein synthesis.

*Reviewer #1:*

The authors describe creation of a transgenic mouse strain permitting cre-dependent expression of a mutant tRNA synthetase from the ROSA26 locus. The transgene permits BONCAT and FUNCAT labeling of newly synthesized proteins in specific cell types. Using this strain, they demonstrate a change in the synthesis of 167 proteins in the hippocampus following a spatial learning paradigm. Although there is a long history of showing that hippocampal learning depends on synthesis of new proteins this is one of the first attempts to provide the list of which new proteins contribute.

Despite significant enthusiasm for the overall goal, for the tool the authors have generated and for some aspects of their experiments, I am concerned about one aspect of their experimental design and one aspect of their statistical analyses.

The design compares animals that learn to avoid a shock in a spatial region with yoked control animals that receive the same shocks but cannot (by design) learn to avoid them. This is in many respects a wonderful control for the stress of animals receiving shocks. However a weakness of the design is that it may actually be much more stressful to receive repeated shocks that one cannot avoid by altering behavior, than to receive the same number of shocks but which one comes to avoid by altering one's behavior. If this is true, the change in protein synthesis could either be due to the learning or could reflect the heightened stress in the control animals. An additional set of untrained controls would resolve this.

The statistical concern has to do with the multiple comparisons that come from simultaneously examining many proteins. The authors do not use a threshold for fold-change and do not do any correction for multiple comparisons across proteins.

*Reviewer #2:*

Here the authors use non-canonical amino acid labeling, proteomics, immunohistochemical labeling and Western blots to identify changes in the makeup of newly-synthesized proteins in the mouse hippocampus following learning paradigm ('accelerated' active place avoidance). They first establish the learning paradigm. They then demonstrate that incorporation of AHA in hippocampi of trained mice is greater than that observed in non-trained mice. They then describe and characterize a novel mouse strain (RC3 mice) that conditionally expresses a mutant tRNA synthetase. These are crossed with CaMK2a-Cre mice resulting in selective expression of mutant tRNA synthetase in the hippocampus. They then compare the population of newly synthesized proteins in trained and non-trained mice by labeling with ANL and analyzing the ANL-containing proteins by FUNCAT, BONCAT, MS and Western blots. They report training associated changes in the synthesis of 167 proteins, which are clustered into several groups; changes for a subset of these were then verified using western blots.

The approach taken here is valid and has the potential to address many interesting and open questions related to changes in protein synthesis associated with a variety of phenomena, in this case, training on a specific task. As far as I can tell, the behavioral tasks were performed appropriately and the controls are in place. My enthusiasm is dampened, however, by the superficial analysis and treatment of the results. Consequently, the findings, as they stand, are not particularly insightful or useful. Having said this, this can be remedied by 1) deeper and better analyses and; 2) contrasting the findings with the literature.

1) The bioinformatics analyses and conclusions are rather superficial and not very convincing. First, attribution of 'change' is based on the statistical significance of a paired Students t-test (4 pairs), regardless of the magnitude of the change. Thus, for about half of such proteins, the measured change was less than 30%. Such low change thresholds are uncommon in such studies as they are rarely reliable. A reasonable way to set a threshold would be to estimate the variance within each group, and set a threshold based on, say, 2 standard variations of the internal variance. Second, the clustering into functional groups (the four distinct clusters mentioned) is questionable. For example, one of these is 'synaptic vesicle recycling'. This cluster contains only 3 protein (complexes): the vacuolar ATPase, WD Repeat Domain 7, and Dmx Like 2. These proteins have been associated with synaptic vesicles but not exclusively so, thus referring to this tiny group as a 'synaptic vesicle recycling' cluster is not justified. Third, proteins that exhibited the strongest changes are kind of ignored (for example a sodium-coupled bicarbonate transporter, 5-fold increase), or Glyoxalase I (14-fold change).

2) The details on the 167 proteins identified are scant. No table is provided in the main text that lists these proteins, their common names, the degree of change following training, the statistical significance of the change, etc. Such a table is provided as supplementary data, but no details are provided to explain it (for example, what does 'NumberPeptides' mean? How can 8 measurements be made from 1 peptide?). This being so, it is difficult to appreciate what proteins change beyond those mentioned explicitly in the text.

3) The literature on protein synthesis and learning is vast. How do prior expectations and findings compare with those described here? Given the importance attributed to local protein synthesis in synaptic plasticity, are any relationships observed between the proteins identified here and dendritic (and axonal) mRNAs identified so far? Along these lines, were postsynaptic proteins and receptors identified (I noted, for example that newly synthesized GluA1 was markedly increased in 3 of the 4 experiments)? Presynaptic proteins? (I noted changes in Synataxin 1B, Munc-18-1, Α-SNAP). Were any of the identified proteins unique in the sense that they were synthesized only after training (so called 'plasticity proteins')? This is not to criticize the veracity of the findings, but contrasting the findings with prior studies and common hypotheses on protein synthesis and memory would help clarify the insights the study provides beyond a catalog of protein names. Unfortunately, the Discussion is not enlightening in this regard as it is, for the most part, a reiteration of the results.

4) Figure 2F is rather surprising as it suggests that half of newly synthesized protein is lost within 8 hours (from 16 to 24 hours). How do such rapid degradation rates compare with the long half-lives (many days and even weeks) reported for most neuronal proteins (e.g. Fornasiero et al., 2018, PMID30315172)? What might be the explanation?

*Reviewer #3:*

Summary:

The authors analyze the newly-synthesized proteome of hippocampal neurons using ANL and SWATH; they investigate changes in treated/untreated animals in an active place avoidance test. They identified 700 proteins in total, observing differential expression in 167. To validate their findings, they used BONCAT-WB finding similar trends in most candidate proteins.

Scope:

LTM consolidation is of general interest. Cell-type specific proteome labeling using ncAAs has been described previously, with a much higher overall proteome coverage than reported here. From a neurobiological point of view, the topic is potentially interesting but enthusiasm is diminished by the very low proteome coverage and the absence of many important controls. Hopefully the authors conducted the control experiments but just left them out of the current manuscript. There are several major issues/suggestions, as outlined below.

1) Does long-term memory in the active place avoidance task require protein synthesis? This is an important experiment to include since it would establish that the changes in protein synthesis observed are important for the long-term change in behavior.

2)The authors are using a new mouse line that they created using CRISPR/CAS to mutagenize the methionyl t-RNA synthetase (NLL-MetRS). There is no characterization of the mouse line in the paper. While the crossed mouse uses a well-established CRE driver, each new line can be slightly different, and ectopic expression of the CRE drivers is not unusual. It is thus highly recommended to include least one staining showing the co-localization of the GFP with a neuronal marker. (In Figure 2E the authors report that GFP is expressed in neurons, but no neuronal marker is shown, just a region rich in neurons and ectopic expression cannot be evaluated.

3) Re: proteomics. There were no controls for the purification of newly-synthesized proteins included in the manuscript. Have these purification tests been done? Data should be included that compares non-specific protein absorption in control samples (from e.g. methionine-injected mutant mice or ANL injected wild-type mice, but ideally both) to specifically purified newly-synthesized proteins. This is an essential point that needs to be addressed as the purification of proteins always results in some background proteins associated with the purification procedure (stickiness of the beads). In the present manuscript, all differentially regulated proteins described in the Results section could be derived from non-specifically adsorbing proteins also from other cell types. The authors need to show their enrichment efficiency and thresholds for accepting proteins for inclusion.

4) Re: proteomics: only a very limited number of proteins were identified and quantified, likely representing the "tip of the iceberg" of the cellular proteome. Previous studies showed more than triple the numbers described in this manuscript. Can the authors comment on their relatively low protein numbers?

5) Also, data upload to an online repository is common practice nowadays, and its commendable that the authors plan to do this. PRIDE offers anonymous reviewer accounts after uploading- this allows the possibility that the reviewers can look at the data.

6) It is very interesting and surprising that the authors report a global change in protein translation 16h after a relatively short learning paradigm. Given the big increase in translation that is shown in the FUNCAT experiment (Figure 1E), it is quite surprising that the final number of differential expressed proteins is only 167, and in this regulated pool only 99/167 increased expression, while the remainder showed a decrease in expression. This regulation seems inconsistent with the global FUNCAT measurement. The proteome of single cells is estimated to include ~ 5000 proteins- making it difficult to understand how regulation of 167- with about half going up and half going down- would have a visible effect on the total visualized new proteome. How do the authors explain this?

7) How were the experimental and control MS samples handled? were the 4 control mice samples obtained at the same time, clicked and then analyzed at the same time as the experimental samples (or with interleaving of experimental and control samples)?

8) Data analysis: The authors should provide an analysis of the overlap in replicates for the MS experiments with ANL labelled proteins and the negative control.

9) How were the MS data normalized?

[Editors’ note: what now follows is the decision letter after the authors submitted for further consideration.]

Thank you for submitting your article "Cell-specific non-canonical amino acid labelling identifies changes in the de novo proteome during memory formation" for consideration by *eLife*. Your article has been reviewed by three peer reviewers, including Sacha B Nelson as the Reviewing Editor and Reviewer #1, and the evaluation has been overseen by a Reviewing Editor and Laura Colgin as the Senior Editor. The following individual involved in review of your submission has agreed to reveal their identity: Erin Margaret Schuman (Reviewer #3).

The reviewers have discussed the reviews with one another and the Reviewing Editor has drafted this decision to help you prepare a revised submission.

Essential revisions:

The statistical issues raised by reviewer 1 and 3 have not been adequately addressed. It is necessary to compute a false discovery rate and to provide the additional replicate information requested by reviewer #3.

Reviewer #1:

The authors have revised their manuscripts and performed some of the requested additional control experiments and analyses. Importantly, they have addressed the key issues of background labeling, the dependence of the learning task on new protein synthesis, and to some degree, the statistical analysis of the data.

I leave it to other more expert reviewers to address the question of whether the issues of background have been adequately dealt with, and to assess whether or not the question of what has been learned from the study has been adequately discussed. The experiments do appear to confirm that the learning requires new protein synthesis. The statistical approach is improved but still lacks adequate control for multiple comparisons. The expected false discovery rate has not been addressed.

In the future it would be helpful for the authors to include the specific changes in the manuscript made with explicit references to where in the manuscript these are to be found.

Reviewer #2:

The authors have addressed the points I raised in my original review. I have no further major comments.

Technology-wise, the approach and genetic model developed will probably be useful for many projects in the future.

As to what has been learned on relationships between protein synthesis and spatial learning, matters remain fuzzy. Perhaps this might be expected in such unbiased studies, and explanatory context might emerge in future studies. Alternatively, perhaps the fuzziness is part of the answer (in analogy to realizations emerging from genome-wide association studies aiming to identify genetic sources of complex traits/diseases).

Time will tell.

Reviewer #3:

The authors have improved the manuscript and now show the data from control experiments. On a positive note, their S/N ratio for ANL-containing proteins compared to PBS-treated control is very good (6986 peptides vs. 207 shared peptides, Figure 3—figure supplement 3A).

Two major remaining points.

Re: statistical analysis of significantly regulated proteins (reviewer 1 comment 2 and reviewer 2 comment 1). The authors have used an absolute fold-change cut-off when the standard in the proteomics field is to use a false-discovery-rate. The papers that the authors cite in their response to the reviewers actually use an FDR- not a simple threshold. I don't understand the reluctance to use an FDR- the Volcano plot looks "ok" (evenly distributed clouds, reasonable shape). The authors should use the FDR method to report statistical significance.

• Biological replicates and experiment numbers: The authors report that they have 3 replicates per experiment (3 mice), but the information on whether the mice belong to the same litter or how many independent experiments were conducted is still missing. This is a key point to understand the power of the data shown. The authors should clearly state litters, biological and technical replicates. Example; In the slice experiment Figure 1E, it is stated that there are 4 mice per experiment, is not clear if they imaged one slice per mouse or more. Again not clear if they are from the same litter and/or experiment. This should be clarified. Related to this- it is desirable if the data uploaded on PRIDE can also be clearly recognized as biological and technical replicates and if the file names used make sense and are easy to cross-walk with the manuscript.

---

## [Author Response]

[Editors’ note: the author responses to the first round of peer review follow.]

[…] In contrast, after discussion, reviewers agreed that characterization of the specificity of the driver mouse could potentially be addressed by referring to other studies. It was also felt that the small number of proteins identified could be discussed without requiring additional experiments. Finally, the concern of reviewer 1 about the possibility of stress-dependent effects rather than learning being required for the changes seen could be addressed by the purity controls and demonstration that the task requires protein synthesis.Reviewer #1:The authors describe creation of a transgenic mouse strain permitting cre-dependent expression of a mutant tRNA synthetase from the ROSA26 locus. The transgene permits BONCAT and FUNCAT labeling of newly synthesized proteins in specific cell types. Using this strain, they demonstrate a change in the synthesis of 167 proteins in the hippocampus following a spatial learning paradigm. Although there is a long history of showing that hippocampal learning depends on synthesis of new proteins this is one of the first attempts to provide the list of which new proteins contribute.Despite significant enthusiasm for the overall goal, for the tool the authors have generated and for some aspects of their experiments, I am concerned about one aspect of their experimental design and one aspect of their statistical analyses.The design compares animals that learn to avoid a shock in a spatial region with yoked control animals that receive the same shocks but cannot (by design) learn to avoid them. This is in many respects a wonderful control for the stress of animals receiving shocks. However a weakness of the design is that it may actually be much more stressful to receive repeated shocks that one cannot avoid by altering behavior, than to receive the same number of shocks but which one comes to avoid by altering one's behavior. If this is true, the change in protein synthesis could either be due to the learning or could reflect the heightened stress in the control animals. An additional set of untrained controls would resolve this.

We share this reviewer’s concern that stress levels may be increased in the yoked compared to learning mice because the former cannot avoid the shocks by altering their behaviour due to learning. However, when we analyse plasma levels of corticosterone, a known stress-related hormone (Gong, et al.,2015), we find that while stress levels were higher in mice which received shocks compared to a naïve non-shocked control group, there was no difference in corticosterone levels between trained and yoked controls (that both received shocks (see updated Figure 1—figure supplement 1). This suggests that the observed changes in protein synthesis are likely due to spatial LTM formation and not heightened stress levels.

The statistical concern has to do with the multiple comparisons that come from simultaneously examining many proteins. The authors do not use a threshold for fold-change and do not do any correction for multiple comparisons across proteins.

We agree with the reviewer that a more stringent statistical analysis is required for our de novoproteomic analysis. In the revised manuscript, instead of using 1D-LC SWATH-MS, we used 2D-LC SWATH-MS which allowed us to identify 1,782 newly synthesised proteins. We used a p value cut-off of ≤0.05 and an absolute fold-change cut-off of ≥1.5, as widely used in SWATH-MS (Wu, et al., 2016, Pascovici et al., 2016). Using these cut-off values, we identified 156 proteins which were significantly altered in synthesis during the 16 hours following training (updated Figure 3D).

Reviewer #2:[…] The approach taken here is valid and has the potential to address many interesting and open questions related to changes in protein synthesis associated with a variety of phenomena, in this case, training on a specific task. As far as I can tell, the behavioral tasks were performed appropriately and the controls are in place. My enthusiasm is dampened, however, by the superficial analysis and treatment of the results. Consequently, the findings, as they stand, are not particularly insightful or useful. Having said this, this can be remedied by 1) deeper and better analyses and; 2) contrasting the findings with the literature.1) The bioinformatics analyses and conclusions are rather superficial and not very convincing. First, attribution of 'change' is based on the statistical significance of a paired Students t-test (4 pairs), regardless of the magnitude of the change. Thus, for about half of such proteins, the measured change was less than 30%. Such low change thresholds are uncommon in such studies as they are rarely reliable. A reasonable way to set a threshold would be to estimate the variance within each group, and set a threshold based on, say, 2 standard variations of the internal variance.

As discussed in our second response to reviewer 1, we agree that a more stringent statistical analysis was required for our de novoproteomic analysis. In the revised manuscript we used a p value cut-off of ≤0.05 and an absolute fold-change cut-off of ≥1.5.

Second, the clustering into functional groups (the four distinct clusters mentioned) is questionable. For example, one of these is 'synaptic vesicle recycling'. This cluster contains only 3 protein (complexes): the vacuolar ATPase, WD Repeat Domain 7, and Dmx Like 2. These proteins have been associated with synaptic vesicles but not exclusively so, thus referring to this tiny group as a 'synaptic vesicle recycling' cluster is not justified.

We agree that the way we initially discussed the clusters may have been misleading. We used cluster names to inform about the possible neuronal function of the proteins in these cluster. Unfortunately, GO analysis and other similar techniques are general tools that are not designed to examine changes in individual cell-types, such as neurons. Also, given that we are examining newly synthesised proteins, rather than the whole proteome, it is unlikely that we will identify a high proportion of any given GO category. Therefore, in the revised manuscript, we used the GO, Reactome, and KEGG databases to manually assign informative names and possible neuronal functions to these clusters. We also clarify how these names were selected in the Materials and methods section of the revised manuscript.

Third, proteins that exhibited the strongest changes are kind of ignored (for example a sodium-coupled bicarbonate transporter, 5-fold increase), or Glyoxalase I (14-fold change).

We agree that the potential role in spatial LTM of individual proteins which displayed large changes in synthesis, such as those pointed out by this reviewer, cannot be disregarded. In our initial analysis we selected candidate proteins using two criteria, fold-change and number of interactions with other regulated proteins. Many of the proteins with the strongest changes, such as those pointed out be the reviewer, did not show evidence of interaction with any other changed proteins, leading us to focus on pathways where multiple proteins were altered in synthesis. However, this lack of observed interaction was likely due to the overly stringent STRING score cut-off used (≥0.7). In the revised manuscript, we used the far more commonly used STRING score cut-off of ≥0.4, resulting in many proteins with large fold-changes being observed in our clusters. Two examples of these are pre-mRNA-processing factor 31 (Prp31) (FC=4.3) and dynactin subunit 3 (Dctn3) (FC=-6.9) (updated Figure 4).

2) The details on the 167 proteins identified are scant. No table is provided in the main text that lists these proteins, their common names, the degree of change following training, the statistical significance of the change, etc. Such a table is provided as supplementary data, but no details are provided to explain it (for example, what does 'NumberPeptides' mean? How can 8 measurements be made from 1 peptide?). This being so, it is difficult to appreciate what proteins change beyond those mentioned explicitly in the text.

We apologise for not clarifying the meaning of the data in the supplementary tables and have fixed this in the revised manuscript. In regards to the reviewer’s question about the number of peptides, we like to clarify that this number represents the number of peptide sequences that were used to identify and then quantify a particular protein. This means that for proteins where the number of peptides was one, one unique peptide sequence was used to identify this protein, with the levels of this peptide being quantified across all 8 samples. We have added this information in the new submission.

3) The literature on protein synthesis and learning is vast. How do prior expectations and findings compare with those described here? Given the importance attributed to local protein synthesis in synaptic plasticity, are any relationships observed between the proteins identified here and dendritic (and axonal) mRNAs identified so far? Along these lines, were postsynaptic proteins and receptors identified (I noted, for example that newly synthesized GluA1 was markedly increased in 3 of the 4 experiments)? Presynaptic proteins? (I noted changes in Synataxin 1B, Munc-18-1, Α-SNAP).

Indeed, in addition to the proteins pointed out by the reviewer, we also observed changes in α- and β-adaptin, as well as PP2A, all of which have been shown to be involved in memory, further confirming the validity of our analysis. We expand on this in the Discussion of the revised manuscript. However, the main advantage of our approach is the nonbiased nature of the de novoproteomic analysis which allowed an identification of both known and potential novel memory-related proteins and mechanisms. One such novel mechanism which is hinted at by our proteomics finding is mRNA splicing, as discussed in detail in the revised manuscript.

Were any of the identified proteins unique in the sense that they were synthesized only after training (so called 'plasticity proteins')? This is not to criticize the veracity of the findings, but contrasting the findings with prior studies and common hypotheses on protein synthesis and memory would help clarify the insights the study provides beyond a catalog of protein names. Unfortunately, the Discussion is not enlightening in this regard as it is, for the most part, a reiteration of the results.

Using SWATH-MS, a “presence vs. absence” type of analysis typically seen in non-quantitative proteomic analysis method is not feasible. We do not discount the presence of proteins which are exclusively synthesised during memory formation. However, in our study, we identified fairly even numbers of proteins that were increased and decreased in synthesis (Figure 3D), which would suggest that instead of merely synthesising specific proteins to enable memory formation, neurons must instead dynamically adjust selected pathways and processes during spatial LTM formation. We discuss this in the tenth paragraph of the Discussion.

4) Figure 2F is rather surprising as it suggests that half of newly synthesized protein is lost within 8 hours (from 16 to 24 hours). How do such rapid degradation rates compare with the long half-lives (many days and even weeks) reported for most neuronal proteins (e.g. Fornasiero et al., 2018, PMID30315172)? What might be the explanation?

In Figure 2F, we do find a 30% reduction in the BONCAT signal between 16 and 24 hours, reminiscent of what we had observed earlier for AHA labelling. Together, this suggests that NCAA availability becomes limited at later time points (Evans et al., 2019). While neuronal proteins have an average half-life of around 4 days, there are many proteins which are degraded much faster (Dörrbaum et al., *eLife,* 2018). It is also likely that these shorter-lived proteins are synthesised more regularly, and as such are more likely to incorporate ANL (Schanzenbächer et al., 2016), which would explain the observed decrease in signal after 16 hours. This bias towards shorter-lived proteins is inherent in all techniques which rely upon protein labelling to examine de novoprotein synthesis.

Reviewer #3:[…] LTM consolidation is of general interest. Cell-type specific proteome labeling using ncAAs has been described previously, with a much higher overall proteome coverage than reported here. From a neurobiological point of view, the topic is potentially interesting but enthusiasm is diminished by the very low proteome coverage and the absence of many important controls. Hopefully the authors conducted the control experiments but just left them out of the current manuscript. There are several major issues/suggestions, as outlined below.1) Does long-term memory in the active place avoidance task require protein synthesis? This is an important experiment to include since it would establish that the changes in protein synthesis observed are important for the long-term change in behavior.

We agree with this reviewer that it is important to confirm that the spatial LTM formation induced by our accelerated APA as behavioural paradigm is dependent on protein synthesis. For the revised manuscript, we therefore administered the protein synthesis inhibitor anisomycin to RC3xCamk2a-Cre mice immediately after training with the 30 minute APA test. In the probe trail 16 hours later, unlike PBS-treated mice, mice that had been administered anisomycin were unable to recall the location of the shock zone (Figure 3—figure supplement 2). This confirms that the spatial LTM formation observed in the 30 minute APA test requires new protein synthesis.

2)The authors are using a new mouse line that they created using CRISPR/CAS to mutagenize the methionyl t-RNA synthetase (NLL-MetRS). There is no characterization of the mouse line in the paper. While the crossed mouse uses a well-established CRE driver, each new line can be slightly different, and ectopic expression of the CRE drivers is not unusual. It is thus highly recommended to include least one staining showing the co-localization of the GFP with a neuronal marker. (In Figure 2E the authors report that GFP is expressed in neurons, but no neuronal marker is shown, just a region rich in neurons and ectopic expression cannot be evaluated.

We agree with this reviewer. In the revised manuscript we have included microscopy images revealing that the NLL-MetRS-EGFP and FUNCAT signal is restricted to hippocampal neurons, using MAP2 as a neuronal marker (Figure 2D).

3) Re: proteomics. There were no controls for the purification of newly-synthesized proteins included in the manuscript. Have these purification tests been done? Data should be included that compares non-specific protein absorption in control samples (from e.g. methionine-injected mutant mice or ANL injected wild-type mice, but ideally both) to specifically purified newly-synthesized proteins. This is an essential point that needs to be addressed as the purification of proteins always results in some background proteins associated with the purification procedure (stickiness of the beads). In the present manuscript, all differentially regulated proteins described in the Results section could be derived from non-specifically adsorbing proteins also from other cell types. The authors need to show their enrichment efficiency and thresholds for accepting proteins for inclusion.

We apologise to this reviewer for not clearly communicating the use of purification controls in our initial submission. We have presented these important controls in the revised manuscript. For all de novoproteomic experiments, we controlled for background-purified proteins using PBS-treated controls. In the revised manuscript, we used 2D/LC MS/MS to identify peptides purified by BONCAT from both ANL- and PBS-treated RC3xCamk2a-Cre mice. Any peptides identified in the PBS-negative control were then excluded from the custom ion library used for SWATH-analysis. In addition to this, we excluded any peptides that were identified in the negative control of our previous BONCAT analysis (Evans et al., 2019). In total, 6,986 peptides were identified exclusively in ANL-treated samples, with 119 peptides being found in both ANL- and PBS-treated samples, and 86 peptides being exclusively found in PBS-treated negative controls (Figure 3—figure supplement 3A). We included this information in the Materials and methods section of the revised manuscript. We also added a list of the peptides found in the experimental and negative control groups to Supplementary file 1.

4) Re: proteomics: only a very limited number of proteins were identified and quantified, likely representing the "tip of the iceberg" of the cellular proteome. Previous studies showed more than triple the numbers described in this manuscript. Can the authors comment on their relatively low protein numbers?

In the revised manuscript we have now used 2D-LC SWATH-MS which increased the number of de novosynthesised proteins to 1,782 (updated Figure 3D). This is similar to the number of proteins identified in other in vivoexperiments which have used ANL labelling (Alvarez-Castelao et al., 2017). We do agree with the reviewer that this likely only represents the “tip of the iceberg”, as in vitroexperiments using primary neurons have identified ≈6,000 proteins (Schanzenbächer et al., 2016). This deviation is likely due to the concentration of ANL being higher in these experiments compared to our in vivolabelling. We have added these considerations to the fourth paragraph of the Discussion.

5) Also, data upload to an online repository is common practice nowadays, and its commendable that the authors plan to do this. PRIDE offers anonymous reviewer accounts after uploading- this allows the possibility that the reviewers can look at the data.

We included the PRIDE repository information in the revised version of the manuscript. The dataset identifier is PXD015820.

6) It is very interesting and surprising that the authors report a global change in protein translation 16h after a relatively short learning paradigm. Given the big increase in translation that is shown in the FUNCAT experiment (Figure 1E), it is quite surprising that the final number of differential expressed proteins is only 167, and in this regulated pool only 99/167 increased expression, while the remainder showed a decrease in expression. This regulation seems inconsistent with the global FUNCAT measurement. The proteome of single cells is estimated to include ~ 5000 proteins- making it difficult to understand how regulation of 167- with about half going up and half going down- would have a visible effect on the total visualized new proteome. How do the authors explain this?

As pointed out by this reviewer, FUNCAT analysis revealed an overall increase in newly synthesised proteins during the 16 hour after training. This was also confirmed by BONCAT-WB, by using the total protein stain REVERT (Figure 5). In the revised manuscript, we used a more stringent statistical cut-off for our BONCAT-SWATH-MS de novoproteomics analysis and identified 156 proteins, which were significantly altered in synthesis during spatial LTM formation (Figure 3D), with 88 proteins being increased in synthesis and 68 proteins being decreased in synthesis.

We do not consider these two methods being directly comparable, because FUNCAT and BONCAT-WB are able to detect and measure the total amount of ANL labelled proteins, whereas BONCAT-SWATH-MS can only compare the relative levels of individual proteins. Therefore, proteins which have a higher initial copy number will have a larger impact on the FUNCAT signal that those with a lower initial copy number, even though BONCAT-SWATH-MS may find a similar alteration in fold-change.

We do, however, find it interesting that we observed many proteins which were decreased in synthesis in our de novoproteomic analysis. As pointed out in the sixth response to reviewer 2, this would suggest that instead of merely synthesising small sets of proteins required for memory formation, neurons may modulate selected proteins and pathways during spatial LTM formation. We discuss this in the tenth paragraph of the Discussion.

7) How were the experimental and control MS samples handled? were the 4 control mice samples obtained at the same time, clicked and then analyzed at the same time as the experimental samples (or with interleaving of experimental and control samples)?

Trained and non-trained mice undertook the 30 minute APA at similar times but on sequential days, in order to rule out any changes in protein synthesis due to differences in circadian cycle. Once mice were sacrificed and proteins extracted in RIPA buffer, the samples were clicked and analysed simultaneously.

8) Data analysis: The authors should provide an analysis of the overlap in replicates for the MS experiments with ANL labelled proteins and the negative control.

As pointed out in our response to comment 3, in the new submission we have provided a list of the peptides that were identified in the experimental and negative control samples.

9) How were the MS data normalized?

In regards to the SWATH-MS data, they were normalised by log transforming the protein peak area and normalising this to the total protein peak area for each run, as previously described by us (Evans et al., 2019).

[Editors' note: the author responses to the re-review follow.]

Essential revisions:The statistical issues raised by reviewer 1 and 3 have not been adequately addressed. It is necessary to compute a false discovery rate and to provide the additional replicate information requested by Reviewer #3.Reviewer #1:The authors have revised their manuscripts and performed some of the requested additional control experiments and analyses. Importantly, they have addressed the key issues of background labeling, the dependence of the learning task on new protein synthesis, and to some degree, the statistical analysis of the data.I leave it to other more expert reviewers to address the question of whether the issues of background have been adequately dealt with, and to assess whether or not the question of what has been learned from the study has been adequately discussed. The experiments do appear to confirm that the learning requires new protein synthesis. The statistical approach is improved but still lacks adequate control for multiple comparisons. The expected false discovery rate has not been addressed.

We agree with this reviewer that, in the proteomics field, it is common to use corrections for multiple comparisons using methods such as the Benjamini and Hochberg procedure. However in our dataset, only 27 proteins remained after adjusting the p value ≤ 0.05. This is common in SWATH-MS studies with low power, such as ours, with multiple testing corrections being recorded to often exclude a high proportion of true positives and thus masking the outcomes of a given study (Ganief et al., 2017; Liu et al., 2019). This notion is further supported by experiments using spiked lysates, which have demonstrated that the corrections can dramatically blunt the SWATH-MS analysis (Wu et al., 2016). This would explain why in our study, we found proteins with adjusted p values > 0.05 (such as NSE, α-adaptin, and AUF1) to be significantly altered in synthesis during spatial LTM formation as shown by western blot analysis (Figure 5). In fact, studies which have directly compared the use of unadjusted and adjusted p-values in the context of cost-prohibitive, low-power proteomic analysis have found that there is little advantage in using FDR-adjusted p-values compared to unadjusted p-values in combination with fold-change cut-offs (Pascovici et a., 2016). We therefore feel that our decision to use unadjusted p-values in combination with a fold-change cut-off is justifiable. For transparency, we have now addressed this issue and included a methodological justification in the revised manuscript (subsection “Bioinformatic analysis SWATH-MS data”).

In the future it would be helpful for the authors to include the specific changes in the manuscript made with explicit references to where in the manuscript these are to be found.

The manuscript went through re-submission and thus, we have treated the manuscript as a new submission, i.e. we have not highlighted the modifications to the original submission. In this resubmission, modifications have been highlighted.

Reviewer #3:The authors have improved the manuscript and now show the data from control experiments. On a positive note, their S/N ratio for ANL-containing proteins compared to PBS-treated control is very good (6986 peptides vs. 207 shared peptides, Figure 3—figure supplement 3A).Two major remaining points.Re: statistical analysis of significantly regulated proteins (Reviewer 1 comment 2 and Reviewer 2 comment 1). The authors have used an absolute fold-change cut-off when the standard in the proteomics field is to use a false-discovery-rate. The papers that the authors cite in their response to the reviewers actually use an FDR- not a simple threshold. I don't understand the reluctance to use an FDR- the Volcano plot looks "ok" (evenly distributed clouds, reasonable shape). The authors should use the FDR method to report statistical significance.

We refer the reviewer to our response to reviewer #1 comment 1.

• Biological replicates and experiment numbers: The authors report that they have 3 replicates per experiment (3 mice), but the information on whether the mice belong to the same litter or how many independent experiments were conducted is still missing. This is a key point to understand the power of the data shown. The authors should clearly state litters, biological and technical replicates. Example; In the slice experiment Figure 1E, it is stated that there are 4 mice per experiment, is not clear if they imaged one slice per mouse or more. Again not clear if they are from the same litter and/or experiment. This should be clarified. Related to this- it is desirable if the data uploaded on PRIDE can also be clearly recognized as biological and technical replicates and if the file names used make sense and are easy to cross-walk with the manuscript.

We have addressed these concerns in our revised manuscript by stating more clearly when technical and biological replicates were used. Regarding the PRIDE repository of our data, we have uploaded more files to this repository to make it easier for readers to use our data.